# Augmenter of liver regeneration regulates cellular iron homeostasis by modulating mitochondrial transport of ATP-binding cassette B8

**Hsiang-Chun Chang, Jason Solomon Shapiro, Xinghang Jiang, Grant Senyei, Teruki Sato, Justin Geier, Konrad T Sawicki, Hossein Ardehali***

Feinberg Cardiovascular and Renal Research Institute, Northwestern University School of Medicine, Chicago, United States

**Abstract** Chronic loss of Augmenter of Liver Regeneration (ALR) results in mitochondrial myopathy with cataracts; however, the mechanism for this disorder remains unclear. Here, we demonstrate that loss of ALR, a principal component of the MIA40/ALR protein import pathway, results in impaired cytosolic Fe/S cluster biogenesis in mammalian cells. Mechanistically, MIA40/ALR facilitates the mitochondrial import of ATP-binding cassette (ABC)-B8, an inner mitochondrial membrane protein required for cytoplasmic Fe/S cluster maturation, through physical interaction with ABCB8. Downregulation of ALR impairs mitochondrial ABCB8 import, reduces cytoplasmic Fe/S cluster maturation, and increases cellular iron through the iron regulatory protein-iron response element system. Our finding thus provides a mechanistic link between MIA40/ALR import machinery and cytosolic Fe/S cluster maturation through the mitochondrial import of ABCB8, and offers a potential explanation for the pathology seen in patients with ALR mutations.

**\*For correspondence:**
h-ardehali@northwestern.edu

## Introduction

Chronic loss of Augmenter of Liver Regeneration (ALR) results in mitochondrial myopathy and cataract, and combined respiratory chain deficiency (*Di Fonzo et al., 2009*; *Calderwood et al., 2016*; *Nambot et al., 2017*). Several mutations in *ALR* have been described for this disorder. Some of these documented mutations result in premature stop codon, while the R194H mutation has been shown to affect the protein stability through increasing dissociation of its cofactor FAD (*Daithankar et al., 2010*; *Nambot et al., 2017*). Cellular studies demonstrated that R194H mutant ALR is not functional, and mutant cells can be complemented by ectopic expression of wild type (WT) ALR (*Di Fonzo et al., 2009*). While patients with *ALR* mutations lack its protein expression, how reduced ALR protein levels are linked to human pathology remains to be determined.

ALR is a sulfhydryl oxidase with two isoforms – a full-length isoform that is predominantly in the mitochondria, but has also been observed in other locations depending on the cell type, and a shorter cytosolic isoform (*Ibrahim and Weiss, 2019*). The cytosolic isoform of ALR, which is thought to function in hepatic regeneration (*Li et al., 2002*; *Ibrahim and Weiss, 2019*), lacks the first 80 amino acids, but retains the catalytic domain necessary to oxidize cysteines. Within the mitochondria, the full-length isoform of ALR localizes to the mitochondrial intermembrane space. Although the function of ALR is not entirely understood, it has been implicated in hepatic regeneration after injury (*Polimeno et al., 2011*), protection against hepatic and renal injury (*Huang et al., 2018*; *Liu et al., 2019*), facilitation of cardiac development (*Dabir et al., 2013*), and maintenance of embryonic stem cells (*Todd et al., 2010*). Mice with liver-specific deletion of ALR from birth develop steatohepatitis and hepatocellular carcinoma and have exacerbated hepatic injury when exposed to

alcohol, while postnatal deletion of ALR in the liver induces oxidative stress and promotes hepatic steatosis (*Gandhi et al., 2015*; *Kumar et al., 2016*; *Kumar et al., 2019*). Previous studies demonstrated that ectopic expression of mitochondria-targeted human ALR rescues the cytoplasmic iron-sulfur (Fe/S) cluster maturation defects in *Erv1*-null yeast (*Lange et al., 2001*), suggesting that ALR may be the functional homolog of yeast Erv1p and play a role in Fe/S cluster biogenesis. However, a recent paper demonstrated that the defects of cytosolic Fe/S clusters in *Erv1*-null yeast may be related to a second mutation leading to decreased glutathione levels (*Ozer et al., 2015*). Therefore, the involvement of yeast Erv1p and mammalian ALR in cytosolic Fe/S cluster biogenesis still requires further investigation.

The synthesis of Fe/S clusters occurs in the mitochondria on the scaffold protein iron-sulfur cluster assembly enzyme (ISCU). Iron destined for Fe/S cluster synthesis is imported into mitochondria by mitoferrin 1/2 and is delivered to ISCU by frataxin. The fully assembled Fe/S clusters are then transferred to Fe/S cluster-containing proteins. Because several cytosolic proteins also require Fe/S clusters for their function, another set of cytosolic proteins, collectively named as the cytosolic iron/sulfur cluster assembly (CIA) machinery, facilitates the incorporation of Fe/S clusters into cytosolic proteins (*Lill et al., 2006*). However, the identity of the molecule that supplies CIA machinery and how it gets out of mitochondria has not been identified. To date, ABCB7 and ABCB8 are the two mammalian mitochondrial transporters thought to be critial for cytosolic Fe/S cluster maturation (*Ichikawa et al., 2012*; *Ye and Rouault, 2010*). Proper cytosolic Fe/S cluster maturation is critical for cell health. Deletion of *Abcb8* in mouse heart leads to cytosolic Fe/S cluster deficiency and spontaneous cardiomyopathy (*Ichikawa et al., 2012*). Conditional deletion of *Iop1*, one component of CIA machinery, in mice results in complete lethality, and deletion of this gene in mouse embryonic fibroblasts leads to cellular iron accumulation and rapid cell death (*Song and Lee, 2011*). Previous studies in yeast have implicated three essential players for cytoplasmic Fe/S cluster biogenesis – Atm1p, Erv1p, and glutathione (*Lill et al., 2006*). Additionally, Erv1 has also been shown to mediate cytosolic Fe/S cluster biogenesis in the parasitic trypanosome *T. brucei* (*Haindrich et al., 2017*).

To support adequate Fe/S cluster biogenesis, iron first enters the cells through transferrin receptor-mediated uptake. It can subsequently be stored in the cytoplasm or incorporated into heme and Fe/S clusters in the mitochondria (*Hentze et al., 2010*; *Ye and Rouault, 2010*). While adequate supply of iron is required for normal cellular function, excess iron can catalyze the formation of reactive oxygen species. Therefore, the levels of cellular iron are tightly regulated. The iron regulatory protein (IRP)-iron response element (IRE) system is the master regulator of cellular iron in mammals. There are two IRPs in mammalian systems, IRP1 and IRP2. In the iron-replete state, IRP1 acquires an Fe/S cluster and functions as a cytosolic aconitase, while IRP2 is proteolytically degraded in an iron- and oxygen-dependent process. In the iron-starved state, IRP1 loses its Fe/S cluster and becomes activated, and IRP2 is stabilized. Both proteins then bind to IREs within the 5'- and 3'-untranslated regions (UTRs) of mRNA and posttranscriptionally regulate gene expression (*De Domenico et al., 2008*; *Hentze et al., 2010*; *Anderson et al., 2012*).

In yeast, Erv1p, together with Mia40, is part of the oxidative folding machinery required for the import of a subset of mitochondrial proteins. Mia40 oxidizes the cysteine residues within imported proteins, and the reduced Mia40 is re-oxidized by Erv1p, effectively recycling Mia40 for further use (*Müller et al., 2008*). A similar electron transfer process has also been demonstrated using recombinant human MIA40 and ALR proteins (*Banci et al., 2011*). A subset of proteins imported by the Mia40/Erv1p machinery, including Tim 13 and Tim 22, are involved in the mitochondrial import of other proteins (*Müller et al., 2008*; *Wrobel et al., 2013*). Nevertheless, the functional significance of this protein import system on cytosolic Fe/S cluster maturation and cellular iron regulation in mammalian systems remains unclear.

In this paper, we demonstrate that cytoplasmic Fe/S cluster maturation depends on a functional mitochondrial protein import system consisting of ALR and MIA40. Mitochondrial ALR facilitates the mitochondrial import of ABCB8, a protein needed for the maturation of cytosolic Fe/S proteins. Thus, a reduction in ALR and eventual reduction in the cytosolic Fe/S clusters leads to activation of IRP1 and subsequent cellular iron accumulation.

## Results

### Downregulation of *Alr* results in cytosolic Fe/S maturation defects

Previous yeast studies demonstrated that loss of Erv1p resulted in cytosolic Fe/S cluster deficiency, which was rescued with forced overexpression of human ALR (*Lange et al., 2001*). Although the defect is restricted to cytosolic protein, humans with chronic loss of ALR demonstrate mitochondrial myopathy with respiratory chain defect (*Di Fonzo et al., 2009*; *Calderwood et al., 2016*; *Nambot et al., 2017*). In MEF, we identified the full-length and short isoforms, with the former one predominantly in the mitochondria and the latter one in the cytoplasm (*Figure 1—figure supplement 1A*), as reported previously (*Ibrahim and Weiss, 2019*). To elucidate the role of ALR in cytosolic Fe/S cluster maturation and mitochondrial function in mammalian cells, we first downregulated *Alr* in mouse embryonic fibroblasts (MEFs) using pooled siRNA and demonstrated a significant reduction of *Alr* at the mRNA level (*Figure 1A*). Of note, three out of the four siRNAs in the pooled siRNA target regions shared by the full-length and the short isoform of ALR, and we observed reduction of protein levels of both isoforms (*Figure 1B* and *Figure 1—figure supplement 1B*). While previous literature in moue embryonic stem cell described significant cell death after *Alr*

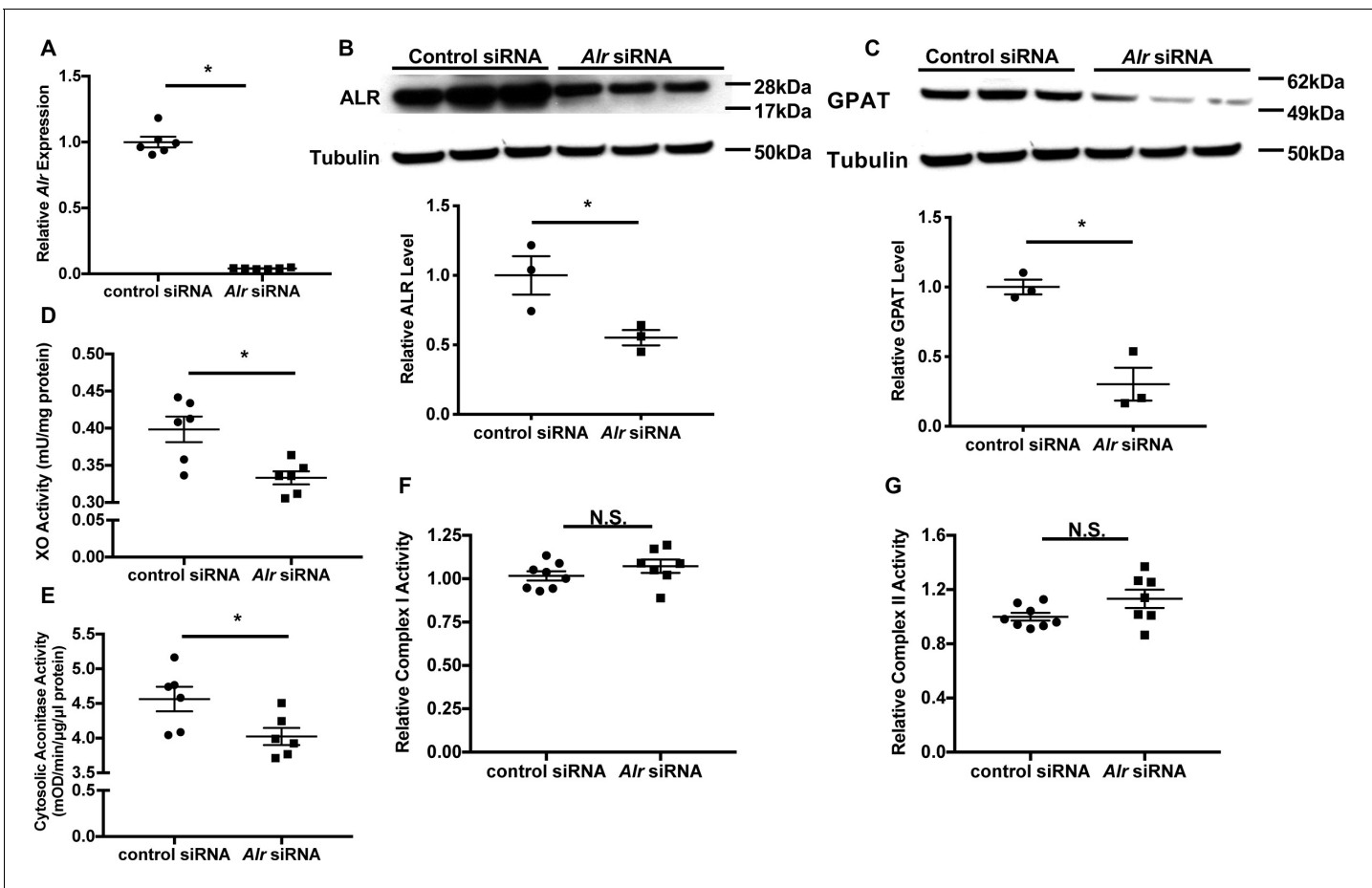

**Figure 1.** Downregulation of *Alr* results in cytoplasmic Fe/S cluster maturation defects. (**A**) mRNA (n = 6) and (**B**) protein (n = 3) levels of *Alr* in wild type (WT) mouse embryonic fibroblasts (MEFs) with or without *Alr* downregulation. GPAT protein levels (n = 3, **C**), xanthine oxidase (XO, n = 6, **D**), and cytosolic aconitase (n = 6, **E**) activities in cells treated with *Alr* siRNA. Mitochondrial complex I (n = 7–8, **F**) and complex II activity (n = 7–8, **G**) in WT MEFs with *Alr* downregulation. Cells were harvested 48 hr after transfection for RNA analysis and 72 hr after transfection for protein analysis. Quantification of western blotting images is shown in the same panel. Data are presented as mean ± standard error mean (SEM). *p<0.05 by ANOVA. N.S. = not significant.

The online version of this article includes the following figure supplement(s) for figure 1:

**Figure supplement 1.** Acute downregulation of *Alr* does not affect cell viability or mitochondrial respiration.

downregulation (*Todd et al., 2010*), we did not observe significant cell death at 72 hr after downregulation of *Alr* (*Figure 1—figure supplement 1C*). This discrepancy likely reflects distinct roles for ALR in differentiated versus undifferentiated cell types. As the quantity of cytosolic Fe/S clusters cannot be measured directly, the protein levels of glutamine PRPP amidotransferases (GPAT), a cytosolic protein whose maturation requires Fe/S clusters, and the activities of xanthine oxidase (XO) and cytosolic aconitase, both of which use Fe/S clusters as cofactors, were used as surrogate markers for cytosolic Fe/S cluster maturation. Downregulation of *Alr* decreased GPAT protein levels (*Figure 1C*), and reduced XO (*Figure 1D*) and cytosolic aconitase activity (*Figure 1E*). We also measured the activity of the Fe/S cluster-containing mitochondrial complexes I and II, and found no change in complex I and II activity with *Alr* downregulation (*Figure 1F and G*). To further assess mitochondrial function, we measured oxygen consumption in intact cells fueled with glucose and L-glutamine using Seahorse Extracellular Flux Assay, and did not observe significant difference in mitochondrial oxygen consumption (*Figure 1—figure supplement 1D*). Our results therefore demonstrated that acute reduction in ALR impairs specifically cytosolic Fe/S cluster maturation in mammalian cells similar to the previously published yeast studies.

## Downregulation of *Alr* increases cellular iron through activation of the IRP-IRE system

We next investigated changes in cellular iron with *Alr* downregulation. Transferrin receptor-1 (*Tfrc*) mRNA levels were increased in wild type MEFs with *Alr* downregulation (*Figure 2A*). Consistent with changes in *Tfrc* expression, we also observed an increase in transferrin-mediated iron uptake in cells treated with *Alr* siRNA (*Figure 2B*). The increased iron uptake resulted in higher steady state levels of iron in both the cytoplasm and mitochondria (*Figure 2C*), similar to other genetic models with cytosolic Fe/S cluster maturation defects (*Sato et al., 2011*; *Ichikawa et al., 2012*). Since non-transferrin-bound iron is not present in normal culturing conditions, these results suggest that downregulation of *Alr* increases cellular iron, likely through upregulation of *Tfrc*.

Both transcriptional and posttranscriptional regulation have been shown to alter *Tfrc* mRNA levels (*Tacchini et al., 2008*; *Anderson et al., 2012*; *Bayeva et al., 2013*). We therefore next studied the mechanism for the upregulation of *Tfrc* by *Alr* knockdown. We first generated a luciferase reporter construct harboring 700 base pairs of the *Tfrc* promoter upstream of the *Tfrc* transcription start site (*Figure 2—figure supplement 1A*). Downregulation of *Alr* did not change luciferase activity of this construct (*Figure 2—figure supplement 1B*), suggesting that ALR does not regulate *Tfrc* expression at the transcriptional level. We next made a construct containing the full-length *Tfrc* 3'-UTR downstream of luciferase cDNA (*Figure 2D*). Luciferase activity of this reporter increased with iron chelation (*Figure 2—figure supplement 1C*), and with *Alr* downregulation (*Figure 2E*), suggesting that ALR posttranscriptionally regulates *Tfrc* through stabilization of *Tfrc* mRNA.

Since *Tfrc* is modulated at the mRNA level by IRPs, we then studied the role of IRP1/2 in ALR-mediated changes in *Tfrc*. We first made a luciferase construct with all IRE sites removed from the *Tfrc* 3'-UTR (*Figure 2D*). As expected, this construct displayed no responsiveness to iron chelation (*Figure 2—figure supplement 1D*). Furthermore, downregulation of *Alr* had no effect on the luciferase activity of the IRE-deleted reporter construct (*Figure 2F*), suggesting that activation of the IRP/IRE system is required for the effects of ALR on cellular iron homeostasis.

We then assessed the role of IRP2 by treating *Ireb2* (encoding IRP2) knockout (KO) MEFs with *Alr* siRNA. *Alr* knockdown in these cells resulted in a significant reduction in ALR protein levels (*Figure 2—figure supplement 1E*), and a significant increase in *Tfrc* mRNA levels and transferrin-mediated iron uptake (*Figure 2—figure supplement 1F and G*), suggesting that the regulation of *Tfrc* by ALR is independent of IRP2. Thus, we focused our studies on IRP1. We first created a stable MEF line expressing *Aco1* (encoding IRP1) shRNA in order to examine the effect of *Alr* downregulation in the absence of IRP1, but we only achieved 40% downregulation of *Aco1* at the mRNA level (*Figure 2—figure supplement 1H*). This could be due to the inefficient expression of shRNA relative to endogenous expression of *Aco1* or the thermodynamic property of the shRNA we utilized. To more effectively downregulate *Aco1* expression, we treated cells expressing *Aco1* shRNA with siRNA against *Aco1* (referred to as *Aco1* KD MEFs hereafter) and achieved a significant decrease in *Aco1* transcript levels (*Figure 2—figure supplement 1I*). Although treatment of *Aco1* KD MEFs with *Alr* siRNA significantly reduced *Alr* mRNA levels (*Figure 2G*), *Tfrc* mRNA levels and TFRC-mediated iron uptake were not increased (*Figure 2G and H*). Since IRP1 is activated upon losing its mature Fe/S

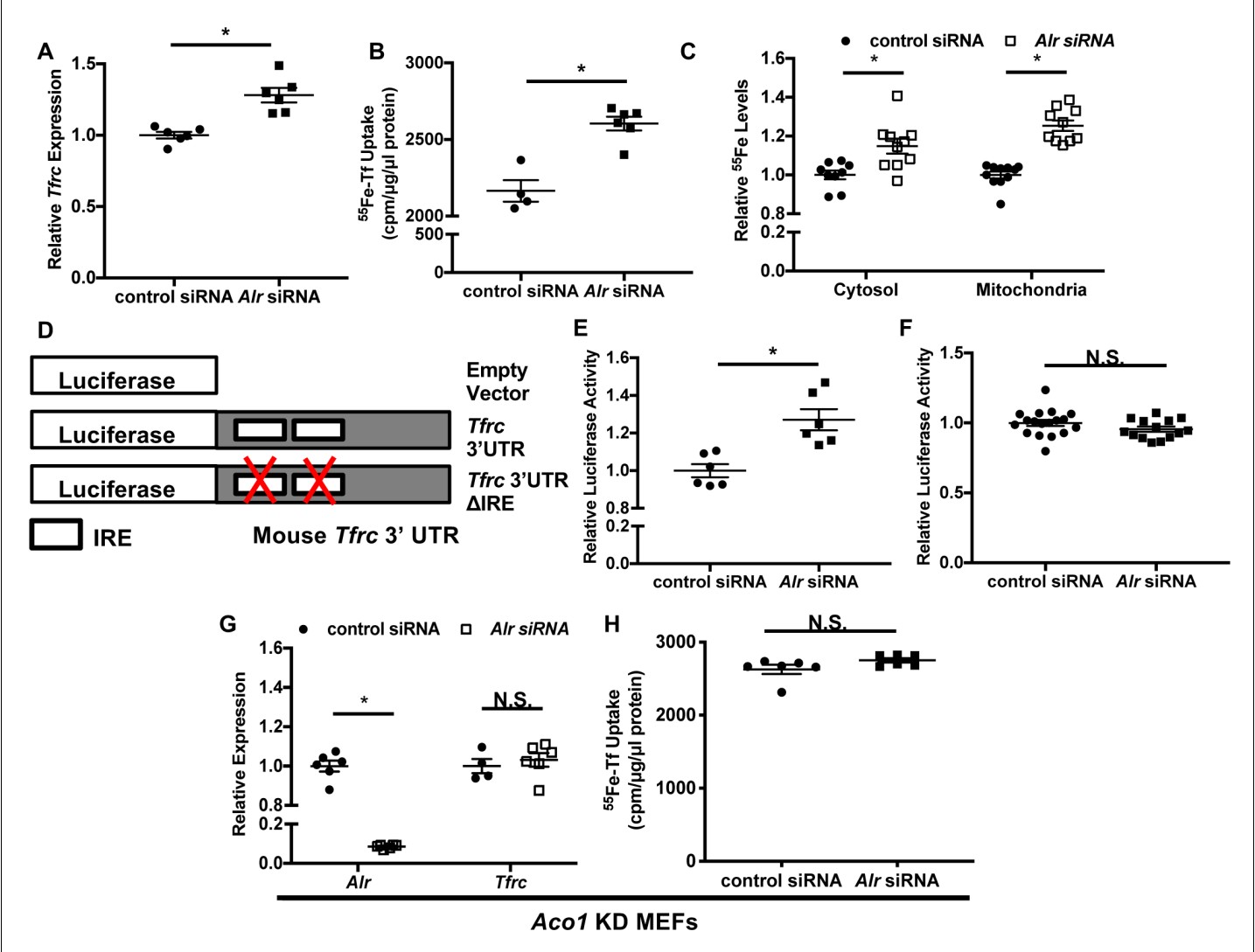

**Figure 2.** Downregulation of ALR increases *Tfrc* levels and cellular iron content through IRP1. *Tfrc* mRNA levels (**A**) and transferrin-dependent iron uptake (**B**) in wild type (WT) mouse embryonic fibroblasts (MEFs) with *Alr* downregulation (n = 5–6). (**C**) Steady-state $^{55}$Fe levels in cytosolic and mitochondrial fraction from cells treated with *Alr* siRNA (n = 6). (**D**) Schematic representation of *Tfrc* 3'UTR reporter constructs. (**E**) Full-length *Tfrc* 3'UTR reporter activities were measured in WT MEFs with ALR downregulation (n = 6). (**F**) IRE-deleted *Tfrc* 3'UTR reporter activities in WT MEFs with ALR downregulation (n = 14–18). (**G**) *Alr* and *Tfrc* mRNA levels in *Aco1* knockdown (KD) MEFs treated with *Alr* siRNA (n = 4–6). (**H**) TFRC-mediated $^{55}$Fe uptake in *Aco1* KD MEFs treated with *Alr* siRNA (n = 6). Experiments were run 48 hr after siRNA transfection for mRNA levels and 60 hr after transfection for steady state $^{55}$Fe levels and transferrin-mediated iron uptake. For luciferase assay, cells were sequentially transfected with siRNA and reporter construct (24 hr apart), and assayed 24 hr after transfection of reporter constructs. Data are presented as mean ± SEM. *p<0.05 by ANOVA. N.S. = not significant.

The online version of this article includes the following figure supplement(s) for figure 2:

**Figure supplement 1.** ALR regulates *Tfrc* mRNA levels through a posttranscriptional mechanism independent of IRP2.

cluster, our results indicate that *Alr* downregulation impairs Fe/S cluster maturation and subsequently promotes increases in *Tfrc* mRNA and cellular iron levels in an IRP1-dependent manner.

## The full-length isoform of ALR regulates cellular iron and cytosolic Fe/S cluster maturation

The ALR gene encodes two isoforms: a full-length isoform predominantly localized to the mitochondria, and a shorter cytoplasmic isoform lacking the first 80 amino acids (*Li et al., 2002*; *Ibrahim and Weiss, 2019*). To determine which isoform of ALR regulates cellular iron and the maturation of

cytosolic Fe/S clusters, we generated lentiviral constructs containing the protein coding regions of the two isoforms of human ALR, but lacking 3'-UTR sequences. Using these constructs, we overexpressed these two isoforms in MEFs and confirmed the localization of each isoform to their respective cellular compartments (*Figure 3—figure supplement 1A and B*).

We then knocked-down endogenous *Alr* with siRNA targeting the 3'-UTR of mouse *Alr* mRNA, followed by lentiviral overexpression of either isoform using the aforementioned virus. This approach allowed us to selectively overexpress individual isoforms of ALR in the absence of the endogenous ALR protein. Quantitative reverse transcription polymerase chain reaction (qRT-PCR) using species-specific primer sets confirmed that only endogenous, but not exogenous, *Alr* was downregulated by siRNA (*Figure 3—figure supplement 1C and D*). Overexpression of the full-length (predominantly mitochondrial) but not the short (cytosolic) isoform of ALR rescued the increase in *Tfrc* mRNA levels and transferrin-dependent iron uptake (*Figure 3A and B*). In addition, the cytosolic Fe/S cluster maturation defects, including reduced cytosolic aconitase and XO activity, and decreased GPAT protein levels, were all reversed by the re-expression of the full-length but not the cytosolic isoform of ALR (*Figure 3C–F*, *Figure 3—figure supplement 1E*). These observations indicate that the full-length ALR isoform regulates cytosolic Fe/S cluster maturation, which in turn affects IRP1 activity and cellular iron levels.

## ALR deficiency impairs mitochondrial transport of ABCB8

Our results thus far indicate that full-length ALR is required for cytosolic Fe/S cluster maturation; however, the mechanism by which ALR carries out this function is not clear. Given that this isoform is predominantly mitochondrial, we first focus our investigation on other mitochondrial proteins implicated in Fe/S cluster biogenesis. ALR downregulation results in Fe/S cluster maturation defects in the cytoplasm but not in the mitochondria, and this phenotype bears striking similarity to cells with deficiency in ABCB7 or ABCB8 (*Pondarré et al., 2006*; *Ichikawa et al., 2012*), two mitochondrial proteins involved in the maturation of cytoplasmic Fe/S clusters. We therefore hypothesized that ALR influences cytosolic Fe/S cluster maturation by modulating the levels of ABCB7 and/or ABCB8. To obtain sufficient quantities of functional mitochondria for further experiments, we used HEK293 cells for the next set of studies. First, we showed that siRNA treatment resulted in a significant decrease in *ALR* mRNA and protein levels in HEK293 cells (*Figure 4—figure supplement 1A and B*). The multiple bands seen in the ALR blot is similar to the previously described molecular weight of full-length ALR in human cells (*Ibrahim and Weiss, 2019*). HEK293 cells with *ALR* downregulation also demonstrated cytosolic Fe/S cluster maturation defects as measured by GPAT protein levels (*Figure 4—figure supplement 1C and D*). However, the mRNA levels of *ABCB7* and *ABCB8* (*Figure 4A*) and total cellular ABCB7 and ABCB8 protein levels (*Figure 4B and C*) did not change with ALR downregulation.

As ALR is involved in mitochondrial protein import, we next examined the amount of ABCB7 and ABCB8 protein contained within the mitochondrial fraction from cells with *ALR* downregulation. *ALR* downregulation did not change the mitochondrial level of ABCB7 (*Figure 4D*), but caused a significant reduction in mitochondrial ABCB8 (*Figure 4E*). Additionally, ABCB7 overexpression did not rescue the increase in *Tfrc* expression in ALR-downregulated MEFs (*Figure 4—figure supplement 1E*), demonstrating lack of functional overlap between ABCB7 and ABCB8. These results suggest that ALR may be required for the mitochondrial import of ABCB8 but not ABCB7.

To determine whether ALR influences the mitochondrial transport of ABCB8, we generated a doxycycline (DOXY)-inducible expression system that allows for monitoring of protein accumulation after DOXY treatment. HEK293 cells stably expressing reverse tetracycline transactivator (rtTA3) were transfected with a tetracycline-responsive bicistronic construct containing GFP and a 6x-His-tagged gene of interest. This system results in mitochondrial protein accumulation in a time-dependent manner in response to DOXY treatment (*Figure 5—figure supplement 1A and B*). To correct for differences in transfection efficiency, the mitochondrial level of the His-tagged protein was normalized to the cytosolic level of GFP.

Mammalian MIA40 does not contain a presequence needed for mitochondrial targeting. Instead, the folding and retention of nascent MIA40 in the mitochondria require a functional oxidative folding system consisting of MIA40 and ALR, both of which are localized to the mitochondrial intermembrane space (*Sztolsztener et al., 2013*). We first validated our assay system by showing that the mitochondrial import of newly synthesized MIA40 was decreased with *ALR* downregulation

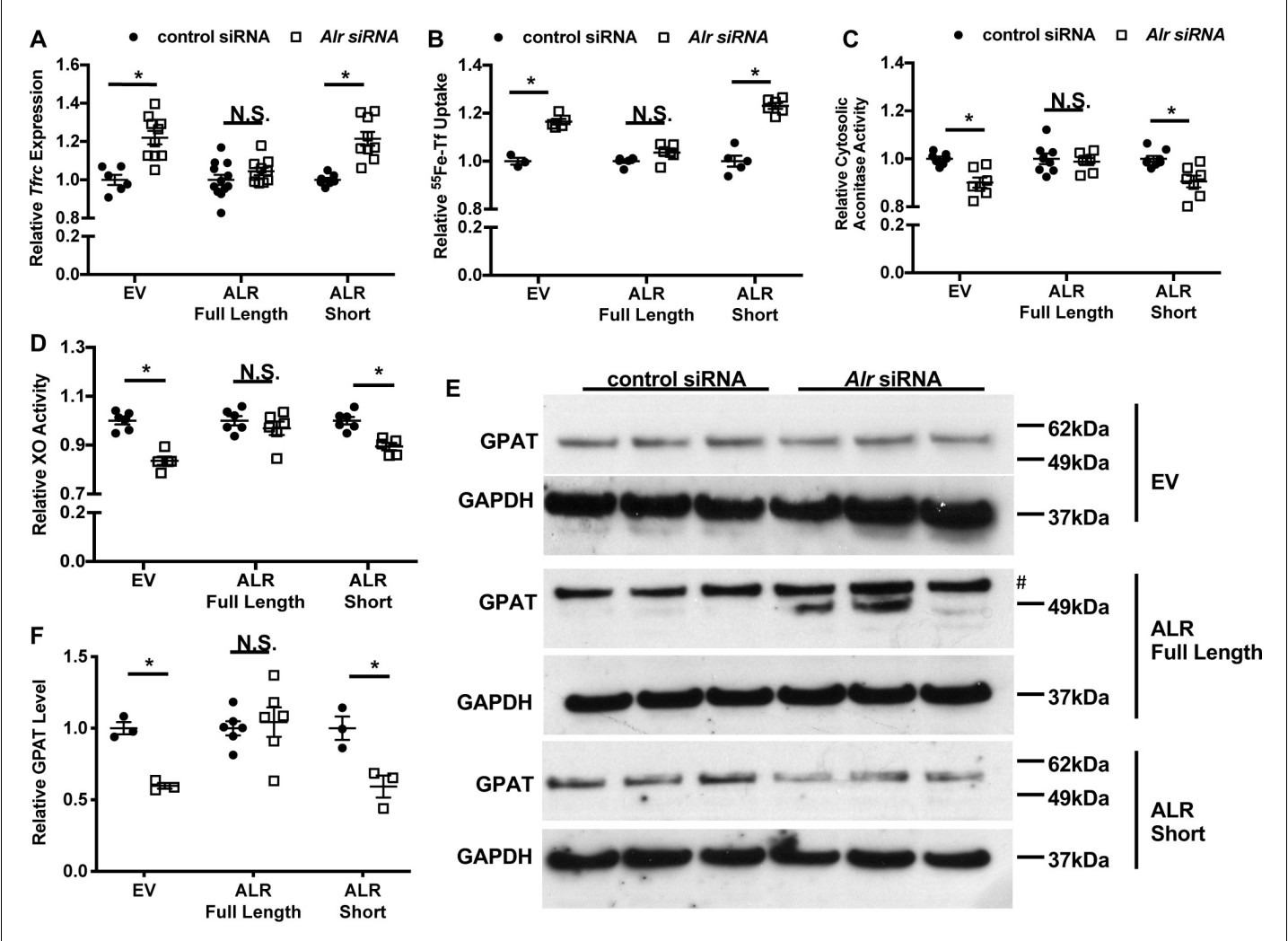

**Figure 3.** Overexpression of full-length but not cytosolic isoform of ALR rescues the iron-sulfur cluster maturation defects from ALR downregulation. *Tfrc* mRNA levels (n = 6, **A**) and transferrin-dependent iron uptake (n = 4–6, **B**) were measured in wild type (WT) mouse embryonic fibroblasts (MEFs) with endogenous *Alr* downregulation and concurrent overexpression of different ALR isoforms. Cytosolic aconitase (n = 7–8, **C**) and XO activities (n = 5–6, **D**), and GPAT protein levels (n = 3–6, **E**) are measured in cells with downregulation of endogenous ALR and overexpression of various ALR isoforms. # indicates the band corresponds to GPAT protein. (**F**) Quantification of panel **E**. EV = empty vector. ALR short = cytosolic isoform of ALR lacking the first 80 amino acids. Data are presented as mean ± SEM. Cells were infected with lentivirus overexpressing the respective ALR isoforms, and transfected with 3'UTR-specific ALR siRNA 24 hr after lentivirus infection. mRNA levels were assayed 48 hr after siRNA transfection, and protein levels and cytosolic XO activity were assayed 72 hr after siRNA transfection. *p<0.05 by ANOVA with post hoc Tukey comparison. N.S. = not significant. The online version of this article includes the following figure supplement(s) for figure 3:

**Figure supplement 1.** Validation of isoform-specific overexpression of ALR with concurrent downregulation of endogenous *Alr*.

(**Figure 5A**). As a negative control, we examined the mitochondrial import of an eGFP fusion protein containing the first 69 amino acids of ATP synthase subunit 9 (SU9), which serves as a mitochondrial localization sequence and mediates mitochondrial import through a membrane potential-dependent but MIA40/ALR-independent mechanism (**Wrobel et al., 2013**). This construct is subsequently referred to as SU9-eGFP. As expected, mitochondrial import of the His-tagged SU9-eGFP did not change with *ALR* downregulation (**Figure 5—figure supplement 1C**). We then studied the effects of ALR downregulation on the mitochondrial import of ABCB7 and ABCB8. *ALR* downregulation resulted in reduced mitochondrial import of ABCB8 (**Figure 5B**), but not ABCB7 (**Figure 5C**). These findings suggest that *ALR* downregulation impairs the mitochondrial import of ABCB8, which in turn affects cytosolic Fe/S cluster maturation.

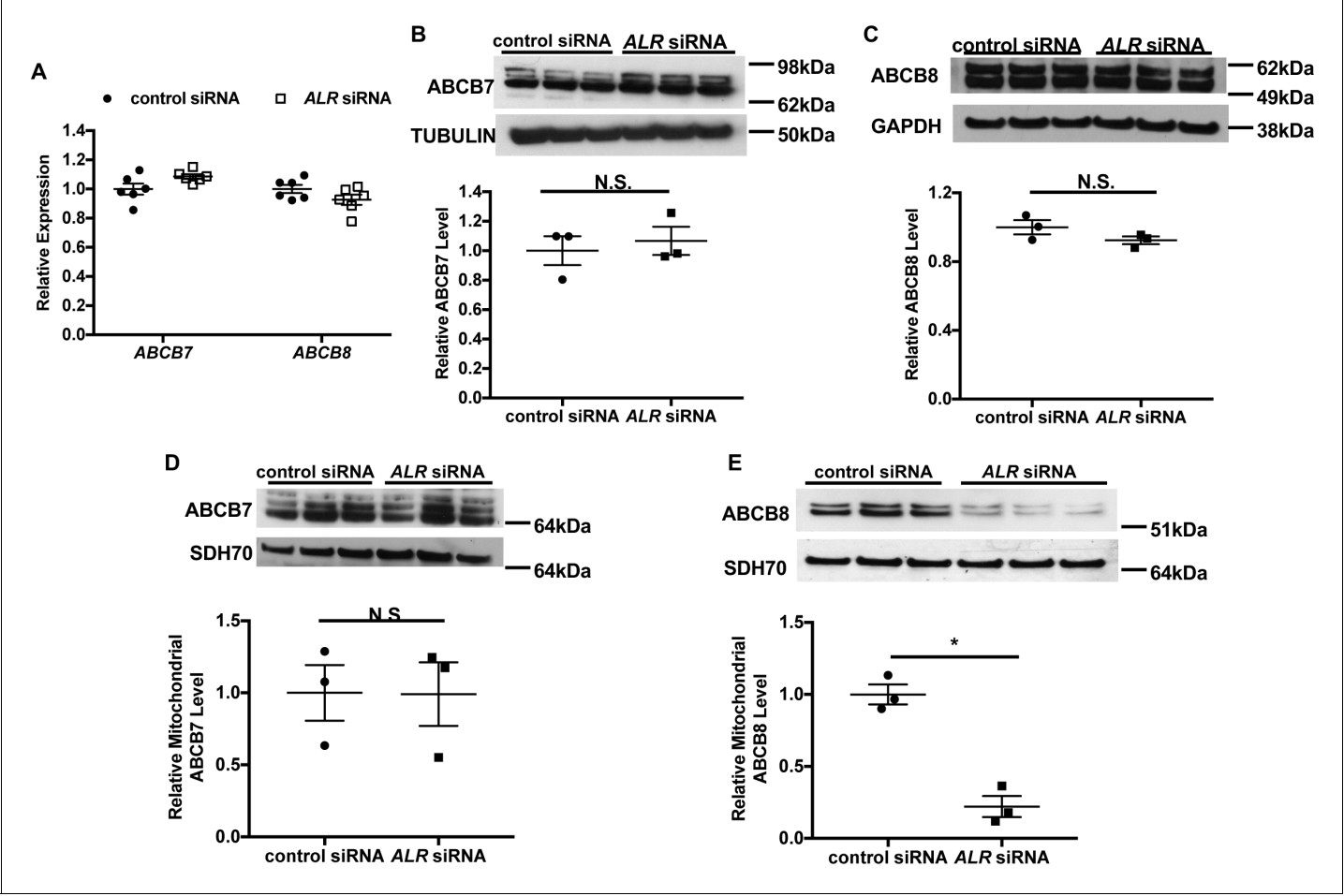

**Figure 4.** Downregulation of *ALR* in HEK293 cells reduces mitochondrial ABCB8 protein level. (**A**) mRNA levels of ABCB7 and ABCB8 in HEK293 cells with *ALR* downregulation harvested 48 hr after siRNA transfection (n = 6). Total cellular levels of ABCB7 (**B**) and ABCB8 (**C**) in HEK293 cells with *ALR* downregulation. Mitochondrial levels of ABCB7 (**D**) and ABCB8 (**E**) in HEK293 cells with *ALR* downregulation. Samples for protein analysis were harvested 72 hr after transfection. Densitometry analysis is shown together with representative images. n = 3 for panels **B–E**. SDH70 = 70 kDa subunit of mitochondrial succinate dehydrogenase, a mitochondrial loading control. Data are presented as mean ± SEM. *p<0.05 by ANOVA. N.S. = not significant.

The online version of this article includes the following figure supplement(s) for figure 4:

**Figure supplement 1.** Downregulation of *ALR* in HEK293 cells results in cytosolic Fe/S cluster maturation defects.

## ABCB8 interacts with MIA40/ALR protein import machinery prior to its transport by the TIM23 complex

While the direct involvement of ALR in mitochondrial protein transport has not been described, ALR is known to play a critical role in the MIA40-mediated protein import pathway by re-oxidizing MIA40 (*Kallergi et al., 2012*). We therefore tested whether downregulation of MIA40 recapitulated the defects of mitochondrial protein transport observed with ALR downregulation. We first confirmed that *MIA40* downregulation in HEK293 cells resulted in increased *TFRC* expression (*Figure 5—figure supplement 1D and E*), similar to what we observed with *ALR* downregulation (*Figure 2A*). In cells with *MIA40* downregulation, we observed a decrease in mitochondrial import of ALR, a known substrate of MIA40 (*Kallergi et al., 2012*), but not SU9-eGFP (*Figure 5D* and *Figure 5—figure supplement 1F*). Similar to our finding with *ALR* downregulation, *MIA40* downregulation impaired mitochondrial import of ABCB8 but not ABCB7 (*Figure 5E and F*). These observations collectively indicate that mitochondrial import of ABCB8 requires both MIA40 and ALR.

Similar to many other inner mitochondrial membrane proteins transported by the TIM23 complex (*Jensen and Dunn, 2002*), ABCB8 also contains a cleavable N-terminal targeting sequence. We

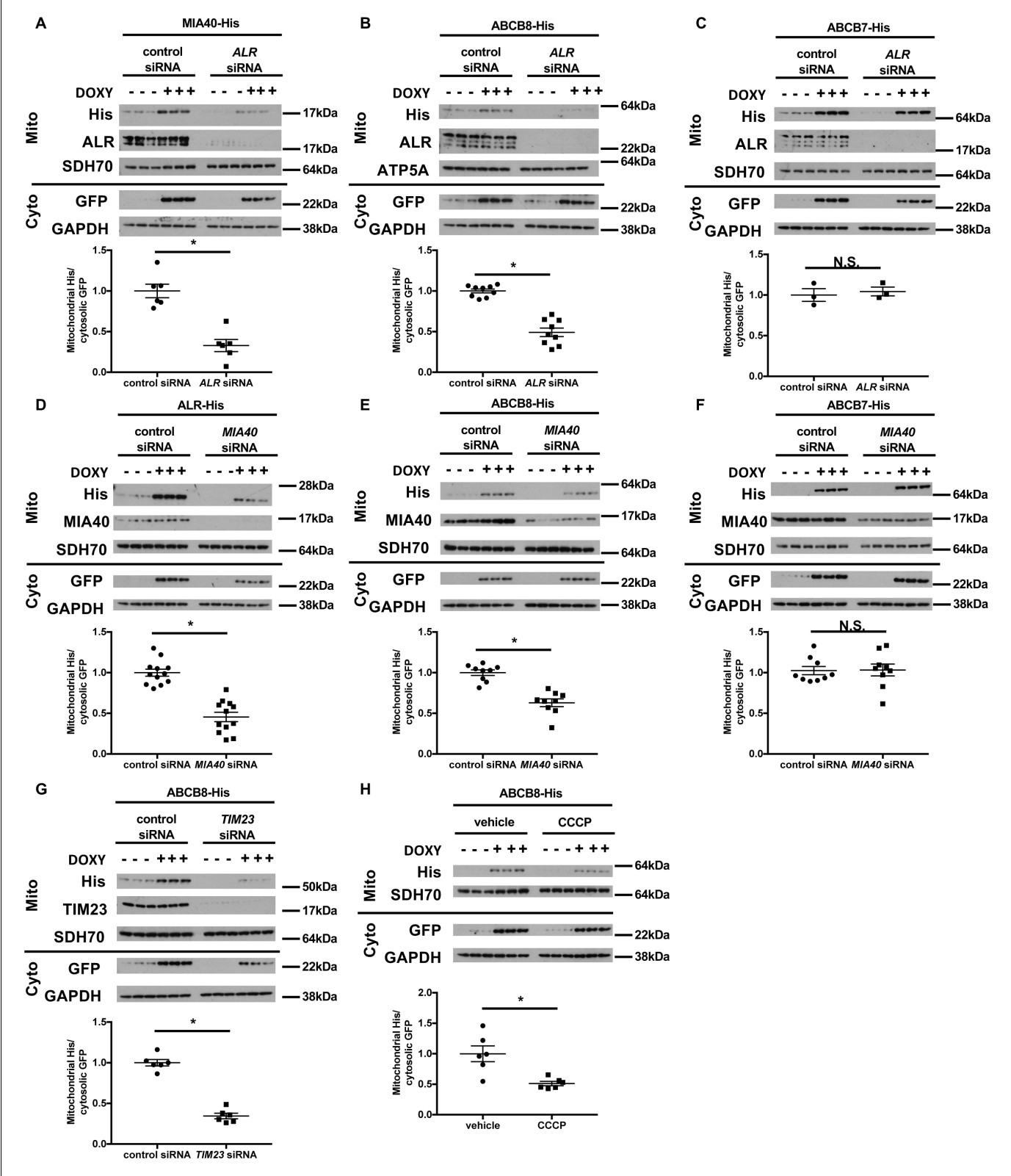

**Figure 5.** Mitochondrial import of ABCB8 requires ALR, MIA40, TIM23, and mitochondrial membrane potential. Imported levels of MIA40-His (n = 6, **A**), whose mitochondrial translocation depends on Mia40/ALR protein transport pathway, ABCB8-His (n = 9, **B**), and ABCB7-His (n = 3, **C**) in cells treated with ALR siRNA with and without doxycycline (DOXY) treatment for 6 hr. Imported levels of ALR-His (n = 12, **D**), ABCB8-His (n = 9, **E**), and ABCB7-His (n = 9, **F**) in cells with or without *MIA40* downregulation with and without DOXY treatment for 6 hr. (**G**) Imported levels of ABCB8-His in cells with or

*Figure 5 continued on next page*

*Figure 5 continued*

without *MIA40* downregulation with and without DOXY treatment for 6 hr (n = 6). (**H**) Imported levels of ABCB8-His in cells treated with carbonyl cyanide 3-chlorophenylhydrazone (CCCP) and/or DOXY (n = 6). Representative western blotting results and densitometry analyses are shown in each panel. Mito = mitochondrial fraction. Cyto = cytosolic fraction. SDH70 = 70 kDa subunit of succinate dehydrogenase. ATP5A = ATP synthase subunit 5A. Cells were first transfected with siRNA and then treated with doxycycline when indicated 48 hr later. Data are presented as mean ± SEM. *p<0.05 by ANOVA. N.S. = not significant.

The online version of this article includes the following figure supplement(s) for figure 5:

**Figure supplement 1.** Downregulation of *ALR* or *MIA40* does not affect mitochondrial import of SU9-eGFP.

therefore tested whether it is also transported by the TIM23 complex. Downregulation of TIM23 (an integral protein of the TIM23 complex) resulted in decreased mitochondrial import of both SU9-eGFP (*Figure 5—figure supplement 1G*, positive control) and ABCB8 (*Figure 5G*), while the mitochondrial import of MIA40 (the negative control) was not affected (*Figure 5—figure supplement 1H*). Additionally, dissipation of mitochondrial membrane potential, a required factor for TIM23-mediated protein transport (*Jensen and Dunn, 2002*), decreased mitochondrial transport of SU9eGFP and ABCB8 (*Figure 5H* and *Figure 5—figure supplement 1I*). These results collectively support a model in which, after passing through the outer mitochondrial membrane, ABCB8 peptide interacts with the MIA40 protein import complex prior to being transported to the inner mitochondrial membrane by the TIM23 complex. A similar mechanism has been described for the yeast mitochondrial ribosomal protein Mrp10 (*Longen et al., 2014*).

To test whether ABCB8 directly interacts with MIA40, we overexpressed C-terminal His-tagged MIA40 in HEK293 cells and loaded the isolated mitochondrial lysate onto a Ni-NTA column to pull down MIA40 and its interacting proteins. Endogenous ABCB8 copurified with MIA40 (*Figure 6A*), suggesting that ABCB8 interacts with MIA40 during its translocation into the mitochondria. Mutation of either or both of the two cysteine residues in MIA40, which have been shown to be required for substrate interaction (*Peleh et al., 2016*), abolished the interaction between MIA40 and ABCB8 (*Figure 6B*).

Since MIA40 recognizes cysteine residues on target proteins to facilitate their mitochondrial import (*Sideris et al., 2009*), we hypothesize that MIA40 and ABCB8 interact through disulfide bond formation. We first performed alignment of ABCB8 protein sequences in vertebrates, and identified five highly conserved cysteine residues (*Figure 6C*). While mutating each of these residues had minimal effect on the interaction between ABCB8 and MIA40 (*Figure 6—figure supplement 1A*), mutation of all five cysteines to serines (5CS-ABCB8) greatly diminished the interaction of ABCB8 to MIA40 (*Figure 6D*). To further support this model, we perform pulldown experiment of wild type MIA40 and ABCB8 in the presence of 100 mM dithiothreitol (DTT). DTT greatly diminished the interaction between MIA40 and ABCB8 (*Figure 6E*), similar to the 5CS mutant. The five conserved cysteine residues are not conserved between ABCB7 and ABCB8 (*Figure 6—figure supplement 1B*), thereby explaining the differential dependency of MIA40/ALR protein import system on mitochondrial import of these two proteins. Thus, these five cysteine residues are collectively needed for recognition by MIA40, and the absence of one is not sufficient to disrupt the interaction between ABCB8 and MIA40. Taken together, these observations suggest that MIA40 and ABCB8 interact through disulfide bond formation.

## Discussion

Mutations in ALR have been linked to mitochondrial myopathy (*Di Fonzo et al., 2009*; *Calderwood et al., 2016*; *Nambot et al., 2017*), but how dysfunctional ALR drives the disease phenotype remained to be determined. Previous studies hint that ALR may function similar to its yeast homolog Erv1p and involve in cytosolic Fe/S cluster maturation (*Lange et al., 2001*). We hereby demonstrated that through its role in mitochondrial intermembrane space, ALR is critical for cytosolic Fe/S cluster biogenesis. The iron overload phenotype of cells with ALR deletion is reminiscent of loss of components of the CIA machinery (*Song and Lee, 2011*) or other mitochondrial protein required for cytosolic Fe/S cluster biogenesis (*Ichikawa et al., 2012*). Importantly, we discovered that downregulation of ALR prevents mitochondrial import of ABCB8, which in turn can result in cytosolic Fe/S cluster maturation defects. Although ALR localizes to various compartments of the

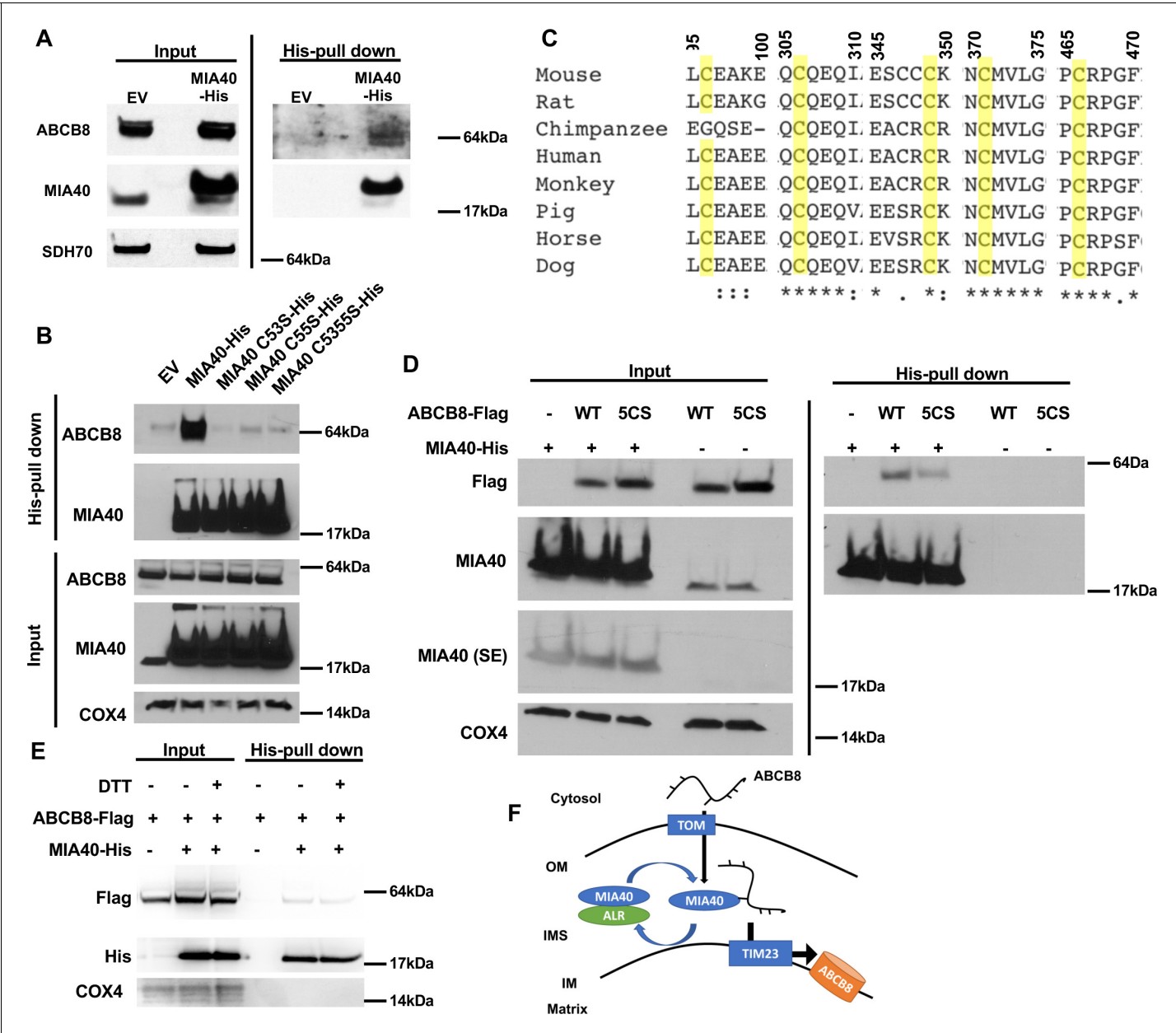

**Figure 6.** ABCB8 interacts with Mia40 during its transport into mitochondria. (**A**) Representative western blot image of mitochondrial lysates and His-tag pulldown fraction from cells overexpressing His-tagged MIA40 or empty vector. (**B**) Co-immunoprecipitation of His-tagged MIA40 and endogenous ABCB8 in mitochondrial fraction from cells overexpressing indicated constructs. (**C**) Clustal Omega alignment of ABCB8 protein sequences from higher vertebrate showing the location of five conserved cysteines (highlighted in yellow). (**D**) Co-immunoprecipitation of His-tagged MIA40 and Flag-tagged ABCB8 in mitochondrial fraction from cells overexpressing indicated constructs. (**E**) Co-immunoprecipitation of His-tagged MIA40 and Flag-tagged ABCB8 in mitochondrial fraction in the presence and absence of 100 mM dithiothreitol (DTT). (**F**) Schematic representation of proposed model in which ABCB8 peptide interacts with the MIA40/ALR protein import machinery prior to insertion into inner mitochondrial membrane through TIM23 complex. EV = empty vector. 5CS = ABCB8 Flag construct with mutations of five conserved cysteines to serine. SE = short exposure. OM = outer mitochondrial membrane. IMS = intermembrane space. IM = inner mitochondrial membrane. WT = wild type.

The online version of this article includes the following figure supplement(s) for figure 6:

**Figure supplement 1.** Individual mutation of the five conserved cysteines in ABCB8 minimally impacts its interaction with MIA40.

cells, our data strongly indicate that the role of ALR in the maturation of cytosolic Fe/S cluster proteins in mammalian systems depends on its localization to the mitochondria and its function in the MIA40/ALR protein import pathway. Our paper thus elucidated the molecular link between ALR and cytosolic Fe/S cluster maturation in mammalian systems.

Patients with mutations in *ALR* that result in reduced ALR protein level develop a syndrome consisting of mitochondrial myopathy, cataract and combined respiratory chain deficiency (*Di Fonzo et al., 2009*; *Calderwood et al., 2016*; *Nambot et al., 2017*). Our mechanistic studies demonstrated that *ALR* downregulation impairs mitochondrial import of ABCB8 protein, and decreased ABCB8 levels have been shown to cause defective cytosolic Fe/S cluster maturation (*Ichikawa et al., 2012*). This decreased mitochondrial ABCB8 level and ensued cytosolic Fe/S cluster maturation defect activate IRP1 and increase cellular iron uptake. Additionally, reduction in mitochondrial ABCB8 levels is also associated with increased sensitivity of cells to oxidative stress (*Ichikawa et al., 2012*). These two mechanisms may explain the profound muscle damage observed in patients carrying *ALR* mutations. It is also worth pointing out that elevated iron level has been causally linked to the cardiomyopathy phenotype in mice with cardiac-specific ABCB8 deletion (*Chang et al., 2016*). Therefore, it would be of great interest to determine genetic or pharmacologic modulation of cellular iron as a therapy for these patients.

Tissue samples from patients with ALR mutation demonstrated depressed mitochondrial respiratory chain activity (*Di Fonzo et al., 2009*; *Calderwood et al., 2016*). In contrast, our data (*Figure 1F and G*) and yeast studies (*Lange et al., 2001*) demonstrated that ALR downregulation does not affect mitochondrial Fe/S cluster assembly, and similar results have been demonstrated with ABCB8 downregulation (*Ichikawa et al., 2012*). There could be two potential explanations for the discrepancy. First of all, not all reported patients with ALR mutations have complex IV deficiency. Therefore, there are likely other disease modifier genes explaining this incomplete penetrance of phenotype. Additionally, the difference between these findings may be due to acute versus chronic depletion of ALR. Chronic mitochondrial iron overload, a cellular response seen in many models of cytoplasmic Fe/S cluster maturation defects (*Lange et al., 2001*; *Pondarré et al., 2006*; *Ichikawa et al., 2012*), has been linked to mitochondrial damage. Extensive mitochondrial damage has also been demonstrated in tissue samples from patients with *ALR* mutations (*Nambot et al., 2017*). It is therefore not surprising that respiratory complex activities will also be reduced in cells full of defective mitochondria. Taken together, respiratory chain deficiency in those patients is likely to be a secondary effect of chronic cytoplasmic Fe/S cluster deficiency.

Mitochondrial proteins utilize different sets of importers to reach their final destination. Import of intermembrane space proteins requires cooperation between the TOM complex and the MIA40/ALR machinery. On the other hand, proper insertion of mitochondrial inner membrane proteins requires both the TOM complex and either the TIM22 or the TIM23 complex (*Webb and Lithgow, 2010*). Both Atm1p, the yeast homolog of ABCB7, and Mdl1p, which shares significant sequence homology with ABCB8, utilize the Tim23 complex for mitochondrial import in a membrane potential-dependent manner (*Webb and Lithgow, 2010*; *Stiller et al., 2016*). In this paper, we demonstrated that the transport of mammalian ABCB8 into the mitochondria is not only TIM23-dependent but also MIA40/ALR dependent. One potential explanation for the dual dependence of ABCB8 on MIA40/ALR protein import machinery as well as TIM23 complex for mitochondrial transport is that MIA40/ALR import machinery is important for the function of TIM23 complex. However, downregulation of MIA40 or ALR does not affect mitochondrial import of the canonical TIM23 complex substrate SU9-eGFP (*Figure 5—figure supplement 1C and D*), and the function of the Tim23 complex was not altered in yeast lacking Mia40 (*Wrobel et al., 2013*). Therefore, modulation of MIA40/ALR complex likely affects the mitochondrial of ABCB8 import upstream of TIM23 and import into the inner mitochondrial membrane. We thus propose a model where a nascent ABCB8 peptide interacts with the MIA40/ALR import system and TIM23 complex sequentially during mitochondrial import, similar to the mechanism that is reported for p53 and Mrp10 for mitochondrial import (*Zhuang et al., 2013*; *Longen et al., 2014*). Taken together, we have identified the MIA40/ALR system as a novel player in mitochondrial import of ABCB8 through direct interaction with the newly synthesized peptide, which occurs prior to peptide insertion into the inner mitochondrial membrane via the TIM23 complex (*Figure 6F*).

Our results demonstrate physical interaction between MIA40 and ABCB8 and suggest that ABCB8 is a novel substrate for the MIA40/ALR protein import system. Our sequence alignment

identifies five cysteine residues that are preserved throughout higher vertebrates. These cysteines likely play a redundant function in the interaction between MIA40 and ABCB8, as the physical interaction between these ABCB8 and MIA40 did not diminish until all five cysteines were mutated. Furthermore, significantly less ABCB8 was copurified with MIA40 under a reducing condition, thereby supporting the model that ABCB8 interacts with MIA40 partially through disulfide bond formation. It is worth noting that the five conserved cysteines, or any other cysteine residues in ABCB8, do not form a $CX_2C$ or a $CX_9C$ consensus motif that is characteristic of most MIA40 substrates (*Sideris et al., 2009*). Therefore, how MIA40 recognizes these residues and how this molecular interaction facilitates mitochondrial transport of ABCB8 require further investigation.

Although our proposed model explains how ALR is involved in cellular iron homeostasis, this mechanism likely presented late in the evolution process. Our multispecies alignment of ABCB8 homologs revealed five conserved cysteines in higher vertebrates, and our experimental evidence argues that all five cysteines are important in the interaction with MIA40. However, not all five cysteines are conserved in lower vertebrates such as zebrafish and *Xenopus*. Additionally, the functional homolog of ABCB8 in yeast has not been identified yet. While it was speculated that Mdl1p was the yeast homolog of mammalian ABCB8 (*Ichikawa et al., 2012*), Mdl1p only shared 28% sequence identity and additional 34% sequence similarity with human ABCB8 protein. Most of the sequence similarities reside in the ATPase domain of the protein. Therefore, our proposed mechanism is likely a late product of evolution and a similar mechanism may not exist in yeast.

In summary, we demonstrated that cytosolic Fe/S cluster maturation in mammalian cells depends on the mitochondrial isoform of ALR, which mediates the transport of ABCB8 into mitochondria. ALR thus plays a role in cellular iron regulation through indirectly regulating IRP1 activity. Sequence variation in conserved cysteine residues in ABCB7 and ABCB8 may explain why the mitochondrial import of only ABCB8 but not ABCB7 depends on the MIA40/ALR pathway. This report presents the first mechanism of how seemingly unrelated mitochondrial proteins involved in sulfur redox homeostasis and mitochondrial protein import are linked together in the cytosolic Fe/S cluster maturation pathway, and provides additional insights into cellular iron regulation, with possible implications in diseases presenting with altered cellular iron homeostasis and Fe/S cluster maturation.

## Materials and methods

### Cloning

*Tfrc* promoter reporter construct was generated by cloning a fragment from −897 bp to +129 bp around mouse TfR1 transcription site into pGL3basic reporter. *Tfrc* 3'UTR luciferase construct was described previously (*Bayeva et al., 2012*). IRE sites in this construct were deleted using Quick-Change Site-Directed Mutagenesis Kit (Agilent). pLenti-rtTA3, SU9-eGFP, and pBI-eGFP plasmids were purchased from Addgene. Human ABCB7 cDNA plasmid was a generous gift from Dr. Berry Paw (Brigham and Women's Hospital and Boston Children's Hospital). Human MIA40 and ALR cDNA clones were purchased from Open BioSystems. Coding sequence of Mia40, ALR, ABCB8, SU9-eGFP, and ABCB7 were tagged with 6x-His tag and subcloned into pBI-eGFP (Addgene) for import experiment. ABCB8 with 6x-His or 3x-Flag tag in the C terminus and Mia40 with 6x-His tag in the C terminus were inserted into pCMV6-XL6. Cysteine mutants of ABCB8 were generated using PCR-based site-directed mutagenesis. All constructs were sequenced before experiments.

### Cell culture

*Ireb2* KO MEFs and wild type MEFs isolated from littermate control were generous gifts from Dr. Tracey Rouault (NICHD, NIH). Mouse embryonic fibroblasts were cultured in Dulbecco's Modified Eagle's Medium (DMEM, Corning) supplemented with 10% fetal bovine serum (FBS), 100 U/ml penicillin, and 100 U/ml streptomycin. We have verified that *Ireb2* was knockout in the *Ireb2* KO MEFs. HEK293 cells were purchased from ATCC and cultured in Minimum Essential Medium (MEM, Corning) supplemented with 10% FBS, 100 U/ml penicillin, 100 U/ml streptomycin, and 1 mM sodium pyruvate. We have not performed separate verification since receiving the HEK293 cell line from ATCC. The cell lines were regularly using PCR-based mycoplasma detection kit (ABM) and were negative for mycoplasma contamination.

## Lentivirus production

Full-length and cytoplasmic isoforms of ALR were cloned into pHIV-eGFP vector. The resultant vector was packaged into lentiviral vector in HEK293T cells after cotransfection with pSPAX and pMD2. G plasmids. *Aco1* shRNA constructs were purchased from Openbiosystems and packaged into lentiviral vector as above. The culture supernatant was mixed with polybrene to a final concentration of 15 µg/ml and used to infect MEFs. To select for cells stably expressing *Aco1* shRNA, MEFs were maintained in complete media with 6 µg/mL puromycin (Sigma) after infection. pLenti-rtTA3 (Addgene) was packaged into lentiviral vector in HEK293T cells after cotransfection with pSPAX and pMD2.G plasmids. Culture supernatant was used to infect HEK293 as described above, and cells with stable integration of viral genome were selected with 10 µg/mL Blasticidin (Invitrogen).

Transfection siRNA against human *ALR, MIA40, TIM23*, and mouse *Alr* and *Aco1* were purchased from Dharmacon. siRNA against 3'UTR sequence of mouse *Alr* were purchased from Qiagen. siRNAs were transfected into HEK293 or MEF using Dharmafect 1 Transfection Reagent (Dharmacon). Plasmids were transfected into MEF using Lipofectamine 2000 reagent (Invitrogen) and into HEK293 cells using either Lipofectamine 2000 reagent (Invitrogen) or calcium phosphate transfection.

## Reverse transcription and quantitative real-time PCR

RNA was isolated from cells using RNA-STAT60 (Tel-Test), and reverse transcribed with Taqman Reverse Transcription Reagents (Invitrogen) according to manufacturers' instruction. Quantification of relative gene expression was done using Fast SYBR Green Master Mix (Applied Biosystems) and run on 7500 Fast Real-Time PCR system (Applied Biosystems). Primer sequences is included in the *Supplementary file 1*.

## Cellular uptake of $^{55}$Fe-transferrin and steady state cellular iron content

$^{55}$FeCl$_3$ (Perkin Elmer) was first conjugated to nitrilotriacetic acid (NTA, Sigma) before adding transferrin (Tf) in 2:1 molar ratio. The mix was incubated at room temperature for one hour. $^{55}$Fe-transferrin was separated from unbound $^{55}$Fe by running on PD-10 desalting column (GE Healthcare). For uptake experiment, cells were incubated with media containing 55 µg/ml $^{55}$Fe-Tf for two hours. Residual membrane-bound iron was removed by washing cells with cold 200 µM deferoxamine (DFO) in phosphate buffered saline (PBS). Radioactivity was quantified on a Beckman scintillation counter, and the results were normalized to cellular protein content of the same sample. For steady state cellular $^{55}$Fe content, cells were incubated with 280 µM of $^{55}$FeNTA for two days. The subcellular fractions were isolated and radioactivity in the fraction was measured as described above.

## Mitochondrial fractioning

Mitochondria from cells were purified using Mitochondrial Isolation Kit for Cells (Pierce) according to manufacturer's instruction.

## Enzyme activity assay

Complex I and II activity were measured with Complex I and Complex II Enzyme Activity Microplate Assay Kit (MitoScience), respectively, according to manufacturer's instruction. Xanthine oxidase activity was measured using AmplexRed Xanthine Oxidase Activity Kit (Invitrogen). For cytosolic aconitase assay, cytosolic fraction was concentrated on Amicon Ultra-0.5 Centrifugal Filter Unit with 10 kDa cutoff (Millipore) and the buffer was exchanged to PBS. The aconitase activity in the concentrated fraction was measured using Aconitase Activity Assay kit (MitoScience).

## Mitochondrial respiration measurement

60 hr after transfections, cells were trypsinized and equal number of cells were replated onto Seahorse XF96 cell culture plates in complete media. Cells were allowed to attach for 8 hr, and then washed with PBS twice to remove the complete media before incubating with assay medium (DMEM without glutamine, with 4.5 g/L glucose, and 2 mM L-glutamine). Oxygen consumption was measured using Seahorse Extracellular Flux Analyzer according to manufacturer's instructions, and mitochondrial oxygen consumption was calculated as baseline oxygen consumption minus non-mitochondrial oxygen consumption (measured in the presence of 50 µM of rotenone and antimycin A).

## Cell death assay

60 hr after siRNA transfection, cells were stained with propidium iodide (PI) and Hoechst 33342. Cell death rate was calculated as the number of PI-positive nuclei over total nuclei number.

## Luciferase assay

Twenty-four hours after transfection with Firefly reporter construct and Renilla luciferase construct (as normalization control for transfection efficiency), cells were solubilized in 1x passive lysis buffer (Promega). The lysates were loaded onto a 96 well plate, and the luciferase activity was determined using Dual Luciferase Assay System (Promega) with Modulus Microplate Reader (Turner Biosystems). The ratio between firefly and renilla luciferase activities was normalized to that of empty vector under corresponding treatments.

## Western blotting

Proteins were resolved on 4–12% Novex Bis-Tris poly-acrylamide gel (Invitrogen) and blotted onto nitrocellulose membrane (Invitrogen). Approximately 20–40 μg of whole cell lysate and approximately 6–10 μg of cytosolic or mitochondrial fractions were loaded onto the gel. Membranes were incubated with primary antibodies for ABCB8 (made in-house), ABCB7, MIA40, ALR, Tim 23, 6x-His, and GFP (ProteinTech), glyceraldehyde 3-phosphate dehydrogenase (GAPDH), ATP synthase F1 subunit alpha (ATP5a), cytochrome oxidase subunit 4 (COX4) and α-tubulin (Abcam), FLAG (Sigma), β-actin (Cell Signaling), GPAT (a generous gift from Dr. Roland Lill [University of Marburg]), or 70 kDa subunit SDH (Invitrogen) overnight, before the addition of horseradish peroxidase (HRP)-conjugated secondary antibodies (Jackson ImmunoResearch). The presence of target protein was visualized using Super Surgical Western Pico ECL substrate (Pierce). Quantification of western blotting image was done using ImageJ (NIH).

## Mitochondrial import of target protein

36 hr after transfection, cells were treated with 1 ug/ml doxycycline, and harvested after 6 hr for mitochondrial fractionation. For carbonyl cyanide 3-chlorophenylhydrazone (CCCP) treatment, 2.5 uM CCCP was added at the fourth hour and the cells were incubated for additional 2 hr. The mitochondrial level of protein of interest is normalized to the GFP levels in the cytosolic fraction from the same sample.

## Ni-NTA pulldown assay

Mitochondrial pallets were lysed in PBS containing 1% Triton X100 (Sigma), protease inhibitor (G Biosciences) and 10 mM imidazole (GE Healthcare). The lysate was cleared via centrifugation and loaded to His Spin Trap spin column (GE Healthcare). The column was sequentially washed with PBC containing 20, 40, and 80 mM imidazole and eluted with PBS containing 200 mM imidazole. 100 mM DTT was included in the lysis buffer when indicated.

## Statistical analysis

Data are presented as mean ± standard error mean (SEM). Statistical significance was assessed with ANOVA, with post hoc Tukey's test for multiple group comparison. All analysis were conducted using GraphPad Prism nine software. A P-value less than 0.05 was considered statistically significant.

## Acknowledgements

We would like to thank Drs. Roland Lill, Barry Paw, Tracey Rouault, and Celeste Simon for sharing reagents. We would also like to thank technical help from Chunlei Chen. H-CC is supported by American Heart Association 12PRE12030002 and NIH T32 GM008152. HA is supported by NIH R01 HL127646, R01 HL140973, and R01 HL138982.

## Additional information

### Competing interests

Hossein Ardehali: Reviewing editor, *eLife*. The other authors declare that no competing interests exist.

### Funding

| Funder | Grant reference number | Author |
|---|---|---|
| NIH Office of the Director | R01 HL127646 | Hossein Ardehali |
| NIH Office of the Director | R01 HL140973 | Hossein Ardehali |
| NIH Office of the Director | R01 HL138982 | Hossein Ardehali |
| American Heart Association | 12PRE12030002 | Hsiang-Chun Chang |
| NIH Office of the Director | T32 GM008152 | Hsiang-Chun Chang |

The funders had no role in study design, data collection and interpretation, or the decision to submit the work for publication.

### Author contributions

Hsiang-Chun Chang, Conceptualization, Data curation, Software, Formal analysis, Validation, Investigation, Methodology, Writing - original draft, Project administration, Writing - review and editing; Jason Solomon Shapiro, Data curation, Validation, Investigation, Methodology, Writing - review and editing; Xinghang Jiang, Teruki Sato, Justin Geier, Investigation; Grant Senyei, Investigation, Methodology; Konrad T Sawicki, Formal analysis, Investigation, Writing - review and editing; Hossein Ardehali, Conceptualization, Formal analysis, Supervision, Funding acquisition, Investigation, Writing - review and editing

### Author ORCIDs

Hsiang-Chun Chang ⓘ https://orcid.org/0000-0002-9201-4500
Jason Solomon Shapiro ⓘ http://orcid.org/0000-0003-0880-3142
Konrad T Sawicki ⓘ http://orcid.org/0000-0003-2124-0081
Hossein Ardehali ⓘ https://orcid.org/0000-0002-7662-0551

### Decision letter and Author response

Decision letter https://doi.org/10.7554/eLife.65158.sa1
Author response https://doi.org/10.7554/eLife.65158.sa2

## Additional files

### Supplementary files

• Supplementary file 1. Primer sequences for quantitative RT PCR. The protocol for qRT-PCR were described in 'Materials and methods' section.

• Transparent reporting form

### Data availability

There are no sequencing or structural data generated in the manuscript. All data generated and analyzed during this study are included in the manuscript and supporting files.

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

# Appendix 1

**Appendix 1—key resources table**

| Reagent type (species) or resource | Designation | Source or reference | Identifiers | Additional information |
|---|---|---|---|---|
| Antibody | Rabbit polyclonal anti-GPAT antibody | Dr. Roland Lill, University of Marburg | N/A | WB(1:1000) |
| Antibody | Rabbit polyclonal anti-ABCB8 antibody | Lab generated. For reference, see Ardehali et al 2004. | N/A | WB(1:1000) |
| Antibody | Rabbit polyclonal anti-GAPDH antibody | Abcam | ab9485 | WB(1:6000) |
| Antibody | Rabbit polyclonal anti-ALR antibody | ProteinTech | 11293–1-AP | WB(1:1000-1:4000) |
| Antibody | Rabbit polyclonal anti-ABCB7 antibody | ProteinTech | 11158–1-AP | WB(1:500) |
| Antibody | Rabbit polyclonal anti-alpha tubulin antibody | Abcam | ab4074 | WB(1:2000) |
| Antibody | Mouse monoclonal anti-SDH70 antibody (2E3GC12FB2AE2) | Invitrogen | 459200 | WB(1:7000) |
| Antibody | Mouse monoclonal anti-6x-His antibody | ProteinTech | 66005–1-Ig | WB(1:4000-1:6000) |
| Antibody | Mouse monoclonal anti-GFP antibody | ProteinTech | 66002–1-Ig | WB(1:4000-1:6000) |
| Antibody | Mouse monoclonal anti-ATP5A antibody [15H4C4] - Mitochondrial Marker | Abcam | ab14748 | WB(1:4000) |
| Antibody | Rabbit polyclonal anti-Mia40 antibody | ProteinTech | 21090–1-AP | WB(1:4000) |
| Antibody | Rabbit polyclonal anti-Tim23 antibody | ProteinTech | 11123–1-AP | WB(1:4000) |
| Antibody | Mouse monoclonal Flag M2 antibody | Sigma | F1804 | WB(1:1000-1:4000) |
| Antibody | Mouse monoclonal anti-COX4 antibody [1A12A12] | Abcam | ab110272 | WB(1:1000) |
| Antibody | Rabbit polyclonal anti-beta actin antibody | Cell Signaling | 4967 | WB(1:2000) |
| Antibody | HRP-conjugated donkey polyclonal anti-rabbit IgG antibody | Jackson ImmunoResearch | 711-035-152 | WB(1:2000-1:12000) |
| Antibody | HRP-conjugated donkey polyclonal anti-mouse IgG antibody | Jackson ImmunoResearch | 715-035-150 | WB(1:2000-1:12000) |
| Chemical compound, drug | $^{55}$Fe radionuclide. | Perkin Elmer | NEZ043002MC | |
| Chemical compound, drug | RNA-Stat60 | Teltest | Cs-502 | |

*Continued on next page*

*Appendix 1—key resources table continued*

| Reagent type (species) or resource | Designation | Source or reference | Identifiers | Additional information |
|---|---|---|---|---|
| Chemical compound, drug | Glycogen | Life Technologies | AM9510 | |
| Chemical compound, drug | Lipofectamine 2000 | Life Technologies | 11668019 | |
| Chemical compound, drug | Apo-transferrin, human | Sigma | T1147 | |
| Chemical compound, drug | Dharmafect Transfection Reagent I | Dharmacon | T-2001–03 | |
| Chemical compound, drug | Deferoxamine mesylate | Sigma | D9533-1G | (150 µM) |
| Chemical compound, drug | Doxycycline | Fisher Scientific | AC446060050 | (1 µg/mL) |
| Chemical compound, drug | Blasticidin | Life Technologies | R21001 | (10 µg/mL) |
| Chemical compound, drug | Puromycin | Sigma | P8833-25MG | (6 µg/mL) |
| Chemical compound, drug | DMEM, 4.5 g/L glucose with L-glutamine and sodium pyruvate | Corning | 10-013CV | |
| Chemical compound, drug | MEM | Corning | 15-010CV | |
| Chemical compound, drug | Sodium pyruvate solution | Corning | 25–000 CI | |
| Commercial assay, kit | qScript cDNA Synthesis Kit | Quanta | 95047–500 | |
| Commercial assay, kit | PerfeCTa SYBR Green FastMix | Quanta | 95074–05K | |
| Commercial assay, kit | Dual-Luciferase Reporter Assay System | Promega | E1980 | |
| Commercial assay, kit | BCA Protein Assay Kit | Pierce | 23225 | |
| Commercial assay, kit | Amplex Red Xanthine/Xanthine Oxidase Assay Kit | ThermoFisher | A22182 | |
| Commercial assay, kit | Complex I Enzyme Activity Microplate Assay Kit | Abcam | ab109721 | |
| Commercial assay, kit | Complex II Enzyme Activity Microplate Assay Kit | Abcam | ab109908 | |
| Commercial assay, kit | Mitochondria Isolation Kit for Cultured Cells | Thermo Scientific | 89874 | |

*Appendix 1—key resources table continued*

| Reagent type (species) or resource | Designation | Source or reference | Identifiers | Additional information |
|---|---|---|---|---|
| Commercial assay, kit | Aconitase Activity Assay Kit | Abcam | ab109712 | |
| Commercial assay, kit | Seahorse Extracellular FluxPak | Agilent | 102416–100 | |
| Cell line (human) | HEK293 cells | ATCC | CRL-1573 | |
| Transfected construct (human) | HEK293 cells stably expressing rtTA3 | This manuscript. | N/A | Cells were stably infected with rtTA3 lentivirus made from pLenti CMV rtTA3 Blast (w756-1) plasmid |
| Cell line (mouse) | WT and IRP2 KO mouse embryonic fibroblasts | Dr. Tracey Rouault NICHD, NIH | N/A | |
| Transfected construct (mouse) | WT MEFs stably expressing *Irebp1* shRNA | This manuscript | N/A | Cells were stably infected with lentivirus expressing mouse *Irebp1* shRNA made from rtTA3 lentivirus made from pGIPZ-shIrebp1 |
| Sequence-based reagents | siGenome siRNA against mouse GFER | Dharmacon | MQ-065586-01-0002 | |
| Sequence-based reagents | Negative Control siRNA | Qiagen | 1027310 | |
| Sequence-based reagents | siGenome siRNA against human GFER | Dharmacon | MQ-065586-01-0002 | |
| Sequence-based reagents | siGenome siRNA against mouse IRP1 | Dharmacon | MQ-043541-01-0002 | |
| Sequence-based reagents | siRNA against mouse ALR 3'UTR | Qiagen | SI01011199, SI01011220 | |
| Sequence-based reagents | Primer for qRT-PCR (*Supplementary file 1*) | This manuscript | N/A | Please see *Supplementary file 1* |
| Recombinant DNA reagent | pGIPZ-shIrebp1 | OpenBioSystems | V2LMM_82191 | |
| Recombinant DNA reagent | pRL-TK (*Renilla* Luciferase) | Promega | E2241 | Luciferase reporter construct expressing Renilla luciferase |
| Recombinant DNA reagent | pMIR-REPORT | Ambion | AM5795 | Empty 3'UTR luciferase reporter construct expressing Firefly luciferase |
| Recombinant DNA reagent | pGL3 Basic Firefly Reporter construct | Promega | E1751 | Promoter luciferase reporter construct expressing Firefly luciferase |
| Recombinant DNA reagent | Mouse Tfrc 3'-UTR luciferase and mutants | This manuscript and *Bayeva et al., 2012* | N/A | |
| Recombinant DNA reagent | Mouse Tfrc promoter luciferase construct | This manuscript | N/A | |
| Recombinant DNA reagent | pBI-MCS-EGFP | Addgene | 16542 | |
| Recombinant DNA reagent | pLenti CMV rtTA3 Blast (w756-1) | Addgene | 26429 | |

*Continued on next page*

*Appendix 1—key resources table continued*

| Reagent type (species) or resource | Designation | Source or reference | Identifiers | Additional information |
|---|---|---|---|---|
| Software, algorithm | GraphPad Prism | GraphPad | Version 9 | |
| Software, algorithm | ImageJ | NIH | 1.53 c | |

