## [Decision Letter]

**Acceptance summary:**

This manuscript provides important mechanistic insights into how mutations in the Augmenter of Liver Regeneration (ALR) impacts iron handling and thus cellular iron content. This has potential implications for ALR in regenerative biology in other systems.

**Decision letter after peer review:**

Thank you for submitting your article "Augmenter of Liver Regeneration Regulates Cellular Iron by Modulating Mitochondrial Transport of ATP-Binding Cassette B8" for consideration by *eLife*. Your article has been reviewed by 3 peer reviewers, including Carlos Isales as the Reviewing Editor and Reviewer #1, and the evaluation has been overseen by Mone Zaidi as the Senior Editor. The following individual involved in review of your submission has agreed to reveal their identity: Thomas S Weiss (Reviewer #3).

Essential Revisions:

1. While the work is interesting the manuscript suffers from a lack of convincing data with regards to sorting out Alr effects that are driven by the cytoplasmic isoform, versus the mitochondrial isoform. The separation between the different isoforms is shown only from figure 3; therefore, the results shown in figure 1 and 2 may have been interpreted differently if the two isoforms were differentiated. For example, in figure 1 the authors show that downregulating Alr had no significant effect on mitochondrial complex I and II activities, however, since there was no differentiation between mitochondrial and cytoplasmic isoforms, the lack of effect on mitochondrial complexes may imply enrichment of the cytoplasmic isoform. Additional data would be helpful.

2. The authors describe a stable MEF line expressing Aco1 shRNA that they created in order to test the effects of Aco1 downregulation, however, they achieved only 40% downregulation of Aco1 at the mRNA level. Therefore, they add siRNA for Aco1 and achieved greater decrease in Aco1 transcript levels. Although this technique proved to be more efficient, it is recommended that the authors discuss why the shRNA technique, which is usually very efficient, did not work in this case, as well as how combining shRNA and siRNA is applicable in studying ALR disease.

3. Mitochondrial function studies are missing from the paper. The authors mentioned in the result section that their limitation in performing mitochondrial function experiments was due to a difficulty to attain large amounts of mitochondria from MEF cells, as compared to 293 cells, which were used for this purposes. To my knowledge, MEFs are pretty easy in obtaining mitochondria for mitochondrial studies; furthermore, the authors could use more cells or bigger plates to obtain a high amount of mitochondria. Therefore, to my opinion, this explanation for using 293 cells is not valid.

4. In the material and method section as well in the figure caption the authors should present more details about how the experiments are performed enabling scientists to reproduce the data: e.g.

How long were the cells treated with siRNA before harvesting and analysis? Does the cells after ALR siRNA treatment show signs of apoptosis or cell stress as reported for murine embryonic stem cells after 72h of ALR-siRNA treatment (PMID: 20147447)?

How much protein is loaded on the gel?

What are the primer sequences for PCR differentiating endogenous and exogenous ALR (3'-UTR)?

5. Endogenous expression and localization of ALR isoforms in MEFs and HEK293 cells should be shown (western blot range from 10 to 30 kDa). Figure S3A: There are no visible bands in EV lane in cytosolic and mitochondria fractions, which would correspond to endogenous expression.

---

## [Author Response]

Essential Revisions:1. While the work is interesting the manuscript suffers from a lack of convincing data with regards to sorting out Alr effects that are driven by the cytoplasmic isoform, versus the mitochondrial isoform. The separation between the different isoforms is shown only from figure 3; therefore, the results shown in figure 1 and 2 may have been interpreted differently if the two isoforms were differentiated. For example, in figure 1 the authors show that downregulating Alr had no significant effect on mitochondrial complex I and II activities, however, since there was no differentiation between mitochondrial and cytoplasmic isoforms, the lack of effect on mitochondrial complexes may imply enrichment of the cytoplasmic isoform. Additional data would be helpful.

We have included a representative western blot demonstrating the localization of various isoforms of ALR in MEF (Figure 1 figure supplement 1A). We utilized a pool of 4 siRNA to downregulate ALR in MEF cells. Three out of the four siRNA target regions shared by the long and the short isoform of ALR. Therefore, we would expect that the long isoform of ALR would be targeted equally (if not more preferentially) compared to the short isoform of ALR. Although we only showed the change in full length ALR in the main figure, we included an uncropped gel in Figure 1 figure supplement 1B that demonstrated downregulation of the short isoform of ALR after siRNA treatment. We have also revised the manuscript to better highlight the fact that siRNA targets both long and short isoforms of ALR. To further corroborate our model, we demonstrated the defect of mitochondrial ABCB8 import in cells with Mia40 downregulation. Given the similar phenotype between ALR and MIA40 downregulation and the fact that ALR and MIA40 together form one of the mitochondrial protein import systems, we deduce that mitochondrial ALR is critical for ABCB8 import into the mitochondria and therefore regulates cellular iron homeostasis.

2. The authors describe a stable MEF line expressing Aco1 shRNA that they created in order to test the effects of Aco1 downregulation, however, they achieved only 40% downregulation of Aco1 at the mRNA level. Therefore, they add siRNA for Aco1 and achieved greater decrease in Aco1 transcript levels. Although this technique proved to be more efficient, it is recommended that the authors discuss why the shRNA technique, which is usually very efficient, did not work in this case, as well as how combining shRNA and siRNA is applicable in studying ALR disease.

While shRNAs are generally efficient, knockdown efficiency of the commercially available constructs have not been previously validated. It is well-documented that shRNA efficiency can be influenced by multiple factors, including the level of expression, the secondary structure of the shRNA, the location of the targeted sequences, and the exact sequence and thermodynamic property of the shRNA (Li et al., 2007; Taxman et al., 2010). The purpose of combining shRNA and siRNA is to utilize a multiple-targeting strategy to suppress *Aco1* expression, thereby creating an artificial condition to test whether ALR still has an effect on cellular iron homeostasis independent of *Aco1*. We have added discussion in the text.

3. Mitochondrial function studies are missing from the paper. The authors mentioned in the result section that their limitation in performing mitochondrial function experiments was due to a difficulty to attain large amounts of mitochondria from MEF cells, as compared to 293 cells, which were used for this purposes. To my knowledge, MEFs are pretty easy in obtaining mitochondria for mitochondrial studies; furthermore, the authors could use more cells or bigger plates to obtain a high amount of mitochondria. Therefore, to my opinion, this explanation for using 293 cells is not valid.

We have performed additional mitochondrial functional studies in MEFs and did not observe any changes in baseline oxygen consumption with ALR downregulation. The results are now included in Figure 1 figure supplement 1D.

4. In the material and method section as well in the figure caption the authors should present more details about how the experiments are performed enabling scientists to reproduce the data: e.g.How long were the cells treated with siRNA before harvesting and analysis? Does the cells after ALR siRNA treatment show signs of apoptosis or cell stress as reported for murine embryonic stem cells after 72h of ALR-siRNA treatment (PMID: 20147447)?How much protein is loaded on the gel?What are the primer sequences for PCR differentiating endogenous and exogenous ALR (3'-UTR)?

Thank you for raising this issue. Cells were harvested 48 hours after siRNA treatment for RNA analysis and 60-72hr after siRNA treatment for protein analysis. About 20-40ug of whole cell lysate and around 10ug of mitochondrial fraction were loaded to the gel. Additional details are also now included in figure legends. We did not observe significant cell death with ALR downregulation, and the results are now shown in Figure 1 figure supplement 1C. This difference likely reflects the distinct roles of ALR in differentiated versus undifferentiated cell types. The exogenous ALR sequence is human origin, and we have designed primers targeting specifically for human versus mouse ALR mRNA. The primer sequences are included in Supplemental table 1.

5. Endogenous expression and localization of ALR isoforms in MEFs and HEK293 cells should be shown (western blot range from 10 to 30 kDa). Figure S3A: There are no visible bands in EV lane in cytosolic and mitochondria fractions, which would correspond to endogenous expression.

The antibody we used has limited detection of mouse ALR. The lack of signal in Figure S3A was limited by short exposure period and the relatively low protein amount (6ug) per lane loaded onto the gel. We confirmed the localization of ALR in a separate western blot experiment shown in Figure 1 figure supplement 1A.